# Regularized Offline GFlowNets

**Haozhi Wang**[1,2], **Yunfeng Shao**[2], **Jianye Hao**[1,2], **Yinchuan Li**[2*]
[1]Tianjin University, Tianjin, China   [2]Huawei Noah's Ark Lab, Beijing, China

## Abstract

We propose the regularized offline generative flow networks (RO-GFlowNets) that does not rely on online sampling. Since offline datasets usually cannot cover the entire state space, traditional GFlowNets cannot accurately predict the action sampling probability for each state. To address this problem, RO-GFlowNet aims to minimize the flow matching loss while regularizing the distribution distance of policy and offline datasets. Experimental results show that RO-GFlowNets perform well on offline datasets.

## 1 Introduction

Generative Flow Networks (GFlowNets) (Bengio et al., 2021b) aim to complement reinforcement learning (RL) exploration ability with reward-proportional sampling and have been used in molecular generation (Bengio et al., 2021a; Malkin et al., 2022), causal discovery (Li et al., 2022; Deleu et al., 2022), discrete probability modeling (Zhang et al., 2022) and graph neural networks (Li et al., 2023a;b). However, these methods typically rely on online sampling to stabilize the policy optimization process of GFlowNets. Although GFlowNets' objective function allows for offline training (Bengio et al., 2021a), we found that GFlowNets often struggle to generalize to out-of-distribution (OOD) actions when the data is non-uniform, leading to slower convergence speeds.

The problem of OOD generalization is not unique to GFlowNets, and is also a challenge in the field of RL (Levine et al., 2020). Some offline RL techniques, such as behavior regularization (Wu et al., 2019; Kumar et al., 2020; Agarwal et al., 2020) and model-based policy optimization algorithms (Yu et al., 2021; Wang et al., 2021; Yan et al., 2022), have achieved impressive performance using offline datasets. However, the issue of OOD generalization in GFlowNets remains unresolved. In this paper, we propose the Regularized Offline GFlowNets (RO-GFlowNets) algorithm to achieve OOD generalization capability on offline datasets. We utilize the offline action probability density to constrain the action distribution generated by the sampling policy, while optimizing the flow matching loss function. This approach allows the model to increase the penalty on OOD state-action flow functions, thereby accelerating the convergence rate of the model.

## 2 RO-GFlowNets Algorithm

Define $s' = T(s, a)$ and $F(s)$ as state transition and the total flow through $s$, where $a$ is the action. The edge/action flow $F(s, a) = F(s \rightarrow s')$ is defined as the flow through an edge $s \rightarrow s'$. The training process of vanilla GFlowNets requires summing parent and child flows through nodes (states), and can be optimized by the flow consistency equation: $\sum_{s,a:T(s,a)=s'} F(s, a) = r(s') + \sum_{a' \in \mathcal{A}(s')} F(s', a')$, where $r(s')$ and $\mathcal{A}(s')$ are the reward and the feasible action set $\mathcal{A}(s')$ of state $s'$. Consider an offline dataset $\{s, a, r, s'\} \in \mathcal{D}$ sampled from the exploration strategy $\pi_\beta$. Since the rewards of OOD states and actions are not available in $\mathcal{D}$, the generalization performance of the learned policy is poor. Hence, we propose a constrained flow optimization loss:

$$\min_{\pi \in \Pi} \mathcal{L}(\theta) = \min_{\pi \in \Pi} \mathbb{E}_{\mathcal{D}} \left( \sum_{s,a:T(s,a)=s'} F_\theta(s, a) - r(s') - \sum_{a' \in \mathcal{A}(s')} F_\theta(s', a') \right)^2, \qquad (1)$$

---

*Corresponding Author: Yinchuan Li (e-mail:liyinchuan@huawei.com). This work was completed while Haozhi Wang was an intern of the Huawei Noah's Ark Lab for advanced study.

where $\pi(a|s)$ is defined as $\pi(a|s) = \frac{F(s,a)}{F(s)}$ and $\Pi$ is the unknown constrained policy set. In practice, since the exploration policy $\pi_\beta$ is unknown, we cannot directly solve equation 1.

To solve equation 1, we propose an approximation that constrains $\pi$ to be in $\Pi$ with a differentiable constraint based on dual gradient descent, the optimization problem can be formulated as

$$\pi := \arg\min_\pi \mathcal{L}(\theta), \quad \text{s.t. } \mathbb{E}_{s\sim\mathcal{D}}[\text{Dist}(\mathcal{D}(s), \pi(\cdot|s))] \leq \varepsilon, \tag{2}$$

where $\mathcal{D}(s)$ is the action distribution of $s$ in $\mathcal{D}$, $\varepsilon$ is a small positive value and $\text{Dist}$ is the distribution metric function, such as maximum mean discrepancy (Gretton et al., 2012). To constrain the learned policy, we additionally introduce a neural network $G_\phi$ to approximate the action distribution. For each training step $t$, RO-GFlowNets samples a batch of transition data $\{s_i, a_i, r_i, s_i'\}_{i=1}^N \in \mathcal{D}$ to train $G_\phi$ based on $\mathcal{L}(\phi) = \frac{1}{N}\sum_{i=1}^N H(G_\phi(\cdot|s_i), a_i)$, where $H(p,q) = -\mathbb{E}_p \log_2 q$ is the cross-entropy function. Then we train $F_\theta$ by the flow matching loss (Bengio et al., 2021b; Li et al., 2023c) and transform the policy as $\pi(a|s) = F_\theta(s,a)\mathbb{I}(G_\phi(a|s) > \delta)/F(s)$ for sampling, where $\delta$ is the threshold. The overall algorithm is summarized in Algorithm 1.

---

**Algorithm 1** Regularized Offline GFlowNets Algorithm

---

**Input:** dataset $\mathcal{D}$, mini-batch size $N$, threshold $\delta$.
1: **for** iteration $t = 1, \cdots, T$ **do**
2:     Sample mini-batch data $\{s_i, a_i, r_i, s_i'\}_{i=1}^N \in \mathcal{D}$ and update $G_\phi$ by minimizing $\mathcal{L}(\phi)$.
3:     Update $F_\theta$ by minimizing the flow-matching loss.
4:     Update the policy $\pi$ by $\pi(a|s) = F_\theta(s,a)\mathbb{I}(G_\phi(a|s) > \delta)/F(s)$.
5: **end for**
**Output:** Flow function $F_\theta$, the sampling policy $\pi$.

---

## 3 EXPERIMENT RESULTS

We compare the proposed RO-GFlowNets with vanilla GFlowNets (sampled directly from the offline dataset to obtain a near-optimal policy) and a proposed Model-based Offline GFlowNets (MO-GFlowNets) approach, which uses offline data to train a transition model to achieve online sampling and training. To measure the performance of these methods, we consider the empirical $\ell_1$-error and mode found to demonstrate the superiority of the algorithm.

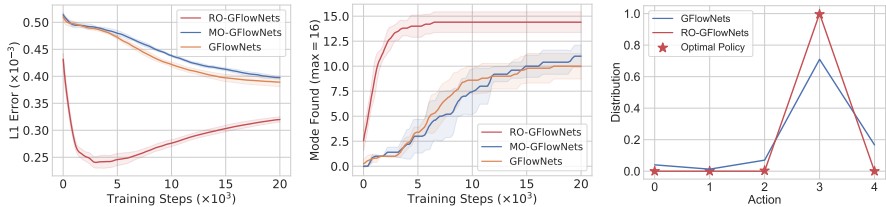

Figure 1: The performance of different algorithms on Hyper-Grid environment. **Left:** $\ell_1$-Error (Lower is better) **Middle:** Mode Found (Higher is better) **Right:** Action Probability Distribution.

Figure 1 demonstrates the performance advantage of RO-GFlowNets, which can find the optimal sampling policy to sample all potential modes from the offline dataset (albeit with a little overfitting), while other algorithms fail. Online sampling training in MO-GFlowNets does not bring gains. This is because the estimated transition model is prone to errors, affecting the optimization of the policy. Figure 1-Right shows the action probability distribution of different algorithms, RO-GFlowNets can fit the optimal policy, which verifies its effectiveness.

## 4 CONCLUSION & FUTURE WORK

We propose an offline training algorithm for GFlowNets, which can avoid the problem of OOD state-action leading to unstable training. Experimental results show that RO-GFlowNets can well learn a sampling policy proportional to the reward function from offline datasets. Future work will be to extend the proposed algorithm to continuous space tasks.

URM STATEMENT

The authors acknowledge that the first and last authors of this work meet the URM criteria of the ICLR 2023 Tiny Papers Track. The first author is 25 years old and non-white student. The last author is 28 years old and non-white.

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

# A  APPENDIX

We first present the experiment details. In the Hyper-Grid environment, the states are the cells of a $N$-dimensional hypercubic grid of side length $H$. The agent starts from the initialization point $x = (0, 0, \cdots)$, and can only increase coordinate $i$ with action $a_i$. In addition, each agent has a stop action. When the agent chooses the stop action or reaches the maximum $H$ of the episode length, the entire system resets for the next round of sampling. The reward function is designed as

$$r(x) = r_0 + r_1 \prod_i \mathbb{I}\left(0.25 < |x_i/H - 0.5|\right) + r_2 \prod_i \mathbb{I}\left(0.3 < |x_i/H - 0.5| < 0.4\right),$$

where $0 < r_0 \ll r_1 < r_2$. The dimension $N$ and length $H$ are set as 4 and 8, respectively. The reward terms $r_0, r_1$ and $r_2$ are set as 1e-5, 0.5 and 2, respectively. $F_\theta$ and $G_\phi$ are optimized by the Adam optimizer with a learning rate 1e-5. Other parameters is same as vanilla GFlowNets.

Figure 2 demonstrates the performance against different thresholds $\delta$. A proper thresholding can achieve good performance on both $\ell_1$-error and mode found criteria. In addition, a higher threshold will reduce performance, because the output distribution of $G_\phi$ will be more uniform at this time, so a higher threshold will make some actions difficult to be sampled.

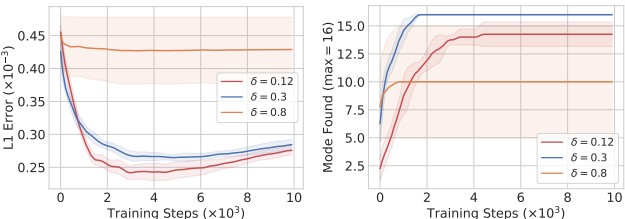

Figure 2: Performance of RO-GFlowNets against different thresholds.

We evaluate RO-GFlowNets on different offline datasets, including the random dataset and random-optimal dataset. The random dataset is sampled from a random policy, and the random-optimal dataset is obtained by using a mixture of optimal and random policies. The optimal policy is obtained by training vanilla GFlowNets until convergence, and the number of offline samples is 20000. Figure 3 shows the comparison results on the random and random-optimal datasets. Since high-reward states are difficult to cover with random datasets, all algorithms find only a few modes. For the random-optimal dataset, the performance of RO-GFlowNets is clearly ahead of other algorithms.

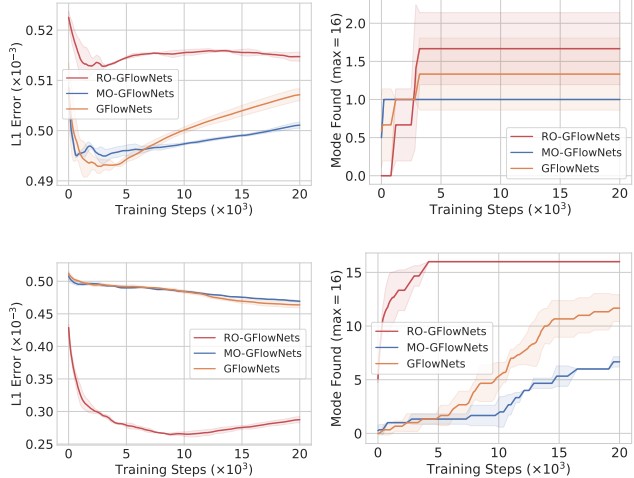

Figure 3: Comparison results on the different datasets. **Upper:** random dataset. **Bottom:** random-optimal dataset.

