# OpenReview forum: "Regularized Offline GFlowNets"
_ICLR.cc/2023/TinyPapers — Submitted to Tiny Papers @ ICLR 2023_

### Official Review · Reviewer_KHhF · 2023-04-01

**Confidence:** 3

**Summary Of Contributions:**

The paper proposes adding a dataset constraint to the regular GFlowNet training objective in order to handle offline training data. The constraint is well-motivated and follows standard the offline RL literature. The resultant objective shows an improvement on a toy example.

**Rating:**

Great Start (GS): a submission which meets some of the reviewing criteria but has room for improvement

**Strengths And Weaknesses:**

Strengths
- Clear presentation and well-motivated addition, resembling the BEAR algorithm in offline RL
- Promising direction for future work, method should scale to larger datasets

**Suggested Changes:**

- Citations should be in parentheses to not break up sentences. E.g. using \citep{...}
- Presentation of Equation 1 is slightly confusing as it is unclear that is constrained (only explained on the next page).

Minor
- ‘To be’ -> ‘to be in’ (page 2 top)

---

### Official Review · Reviewer_8BHD · 2023-04-04

**Confidence:** 2

**Summary Of Contributions:**

The authors propose the Regularized Offline GFlowNets algorithm to solve the problem of OOD generalisation in GFlowNets. They trie to achieve this by regularizing the distribution distance of policy and offline datasets while minimizing the loss. To back their proposed algorithm, they show experimental results that their model works better than vanilla GFLowNets

**Rating:**

Great Start (GS): a submission which meets some of the reviewing criteria but has room for improvement

**Strengths And Weaknesses:**

Strength:
* The paper proposes a novel algorithm to solve the problem of OOD generalisation in GFlowNets and demonstrate effectiveness of their algorithm with ablation against vanilla GFlowNets.

Weakness:
* The paper does not talk about the experimental setup and dataset used to perform the comparison ablation with GFlownet.
* Citations should be in parentheses as they break up the sentences.

**Suggested Changes:**

Please look over the weakness above.

---

### Author Response · Authors · 2023-05-31
**New Paper Revision**

Dear ICLR Program Chairs and Reviewers:

We have uploaded a new revision of our paper. We corrected the unclear formula and added more experimental details

We would like to thank the area chair and reviewers for their time and valuable comments on the paper.

Best regards

---

### Meta-Review · Area_Chair_JoGw · 2023-04-05

**Recommendation:** Invite to present
**Confidence:** 3

**Metareview:**

This paper is clearly written and meets the Clear, Correct, and Reproducible (CCR) standards.

**Summary:**

This paper proposes an offline version of GFlowNet and shows its advantages compared to the vanilla version.

**Reason For Not Giving A Higher Recommendation:**

- More detailed discussion on experimental settings should be provided.
- Somewhat confusing presentation of Eq. 1.

**Reason For Not Giving A Lower Recommendation:**

- This paper tackles an interesting problem.
- This paper is well structured.

---

### Decision · Program_Chairs · 2023-04-07

**Decision:**

Invite to present

**Comment:**

Please add your URM statement.